# Discovery of coding regions in the human genome by integrated proteogenomics analysis workflow

Yafeng Zhu[1], Lukas M. Orre[1], Henrik J. Johansson [1], Mikael Huss[2], Jorrit Boekel[3], Mattias Vesterlund [1], Alejandro Fernandez-Woodbridge[1], Rui M.M. Branca[1] & Janne Lehtiö[1]

Proteogenomics enable the discovery of novel peptides (from unannotated genomic protein-coding loci) and single amino acid variant peptides (derived from single-nucleotide polymorphisms and mutations). Increasing the reliability of these identifications is crucial to ensure their usefulness for genome annotation and potential application as neoantigens in cancer immunotherapy. We here present integrated proteogenomics analysis workflow (IPAW), which combines peptide discovery, curation, and validation. IPAW includes the SpectrumAI tool for automated inspection of MS/MS spectra, eliminating false identifications of single-residue substitution peptides. We employ IPAW to analyze two proteomics data sets acquired from A431 cells and five normal human tissues using extended (pH range, 3–10) high-resolution isoelectric focusing (HiRIEF) pre-fractionation and TMT-based peptide quantitation. The IPAW results provide evidence for the translation of pseudogenes, lncRNAs, short ORFs, alternative ORFs, N-terminal extensions, and intronic sequences. Moreover, our quantitative analysis indicates that protein production from certain pseudogenes and lncRNAs is tissue specific.

[1] Department of Oncology-Pathology, Science for Life Laboratory, Karolinska Institutet, Tomtebodavägen 23A, 171 65 Stockholm, Sweden. [2] Department of Biochemistry and Biophysics, The Arrhenius Laboratories for Natural Sciences, Science for Life Laboratory, Stockholm University, Tomtebodavägen 23A, 171 65 Stockholm, Sweden. [3] Department of Oncology-Pathology, NBIS (National Bioinformatics Infrastructure Sweden), Science for Life Laboratory, Karolinska Institutet, Tomtebodavägen 23A, 171 65 Stockholm, Sweden. These authors contributed equally: Rui M. M. Branca and Janne Lehtiö. Correspondence and requests for materials should be addressed to R.M.M.B. (email: rui.mamede-branca@ki.se) or to J.Lö. (email: janne.lehtio@ki.se)

The impact of genome-level aberrations on the proteome at the systems level is still largely unstudied, especially in organisms with large genomes such as humans. To facilitate such studies, robust methods and workflows that combine sequence data from DNA and RNA analysis with protein-level data are needed. Proteogenomics methods, which combine mass spectrometry-based proteomics data with genomics and transcriptomics data are currently emerging to fill this void[1–3]. Moreover, proteogenomics can be utilized to discover unannotated protein-coding regions both in normal and disease samples. Some coding regions are particularly difficult to annotate correctly without protein-level data, such as translation products from upstream translation initiation sites (TISs) and short open reading frames (sORFs)[4]. Other annotation problems arise when proteins are translated from transcripts that are not expected to be protein-coding, e.g., long non-coding RNAs (lncRNAs) and pseudogenes.

Efficient identification of unannotated coding regions and sequence variants at protein level requires that such variant peptides are included in the database used for mass spectrometry data interpretation. This strategy often leads to a dramatic increase in database size. As an example, a database containing peptides from a six-frame translation (6FT) of the human reference genome is almost 400 times bigger than the database derived from the canonical coding region. A problematic issue in proteogenomics is the accurate estimation of novel peptides' false discovery rate (FDR), especially when large databases are used. This problem is further intensified by the imbalance in probability of correct peptide-spectrum-matching in different search spaces (i.e., canonical search space vs. novel peptide search space) composing the database. In 6FT searches of higher eukaryotic genomes, hypothetical peptides comprise the vast majority of the search space but are actually present in the sample much less frequently than peptides from canonical proteins. Such imbalance can lead to underestimation of FDR with consequences for the sensitivity and reliability of findings[3, 5]. Because of this, 6FT approaches have been rare so far in higher eukaryotes, and instead a more popular strategy has been to concatenate limited sets of putative coding sequences with the canonical protein database. These customized databases are obtained based on data from gene prediction algorithms and other omics techniques, such as genomics, transcriptomics, and ribosome profiling. Using such approach, a number of peptides derived from missense variants (from mutations and non-synonymous SNPs)[6], pseudogenes[7], alternative protein N termini[8, 9], unexpected exon boundaries[10], short open reading frames (ORFs)[11], and alternative reading frame translations (AltORFs)[12, 13] have been identified.

Recently, several bioinformatics tools, CustomProDB[14], Galaxy-P[15], PGTools[16], and JUMPg[17], have been developed to facilitate proteogenomics studies. However, these pipelines mostly resolve issues on generating peptide databases from genomics and transcriptomics data and facilitate visualization of peptide data in the genome scale, not focusing on the curation and validation of the novel findings. Notably, there is an increasing concern about the reliability of reported novel proteins in large-scale proteogenomics studies[3, 18]. In response to this, guidelines for reporting proteogenomics findings have been recommended[19]. Among these, of particular importance are the orthogonal validation by independent methods (e.g., vertebrate conservation analysis, transcriptomics, and ribosome profiling), the special caution that must be devoted to "pseudogene" proteins (at least two high-confidence peptides linked to an in-frame initiation codon required) and the recommendation to discard all novel peptides with single amino acid substitutions[20]. Unfortunately, this latter point results in novel peptides with single substitution being ignored, even though the proteins they originate from could play

important roles in cellular processes. The here presented integrated proteogenomics analysis workflow (IPAW) aims to amend this by offering a reliable solution that allows identification of these single amino acid substituted novel peptides while simultaneously ensuring high confidence of findings by rigorous curation and orthogonal validation.

In this study, we show a proteogenomics workflow integrating discovery, curation, and validation for detecting novel peptides and single amino acid variant (SAAV) peptides. This workflow provides extensive curation steps and allows validation of putative novel peptides by searching for external evidence in orthogonal data. Of particular importance in IPAW is SpectrumAI (Automated Inspection), a tool that curates single amino acid substituted peptides by requiring ions to directly support the residue substitutions in MS/MS spectra. Using a cancer cell line (A431) and five histologically normal human tissues, we generated in-depth proteomics data acquired from an upgraded HiRIEF method[1], which now enables extensive peptide pre-fractionation without bias toward a specific pI range. Applying IPAW to analyze the two data sets, we identify 426 and 155 novel peptides in A431 cells and normal tissues, respectively. Among them, 110 of 117 selected novel peptides are confirmed by synthetic peptide validation. These novel peptides are encoded from a range of supposedly non-coding regions, including pseudogenes, 5′ or 3′ untranslated regions (UTR) of mRNAs, antisense transcripts, dual-coding transcripts, lncRNAs, intergenic, and intronic sequences. Moreover, our quantitative approach with isobaric tags (TMT10plex) shows that several pseudogenes and lncRNAs are translated with tissue specificity; and that several upstream ORF translation products (AltORF and N-terminal extended proteins) differed from the respective canonical protein products in their regulation pattern upon treatment of A431 cells with an EGFR inhibitor (gefitinib). Finally, we find a significantly larger number of novel protein products in A431 cancer cells than in the normal tissues and we note that two types of novel events (retained intron and AltORF proteins) predominantly found in A431 could potentially have stronger immunogenicity due to their distinct long stretches of polypeptide sequence. We demonstrate that the here presented workflow can provide robust data in many proteogenomics applications such as discovery of unannotated coding regions in normal tissues and of mutant peptides specific to cancer cells.

## Results

**Development of full pI range HiRIEF LC-MS**. To increase the potential for discovery of unannotated coding regions in a proteogenomics experiment, in-depth analysis of the proteome is needed. We employed immobilized pH gradient (IPG) strips that cover the full peptide pI range to enable the exploration of the entire tryptic peptidome (Fig. 1a). As a model system, we used the A431 cell line (TMT10-plex labeled peptides of a time course treatment with an EGFR inhibitor, Supplementary Figure 1). The previously established[1] pI range 3.7–4.9 allowed the identification of 93,318 unique peptides when searching against the canonical database. The newly employed pI range 3–10 added 75,825 unique peptides to the results, and further addition of pI ranges 6–9 and 6–11, an additional 34,497 unique peptides (Supplementary Figure 2). Since the combination of pI ranges 3.7–4.9 and 3–10 achieved the bulk of the analytical depth, we applied only these two strips in a subsequent analysis of the normal tissue samples (Supplementary Figure 1). A total of 203,640 and 169,471 peptides were identified (peptide level FDR 1%) from the A431 and normal tissues experiments, respectively, corresponding to 10,166 and 10,889 genes (protein-level FDR 1%) (Table 1). Combining peptides identified from the two experiments, we covered half of the fully tryptic human peptides existing in the

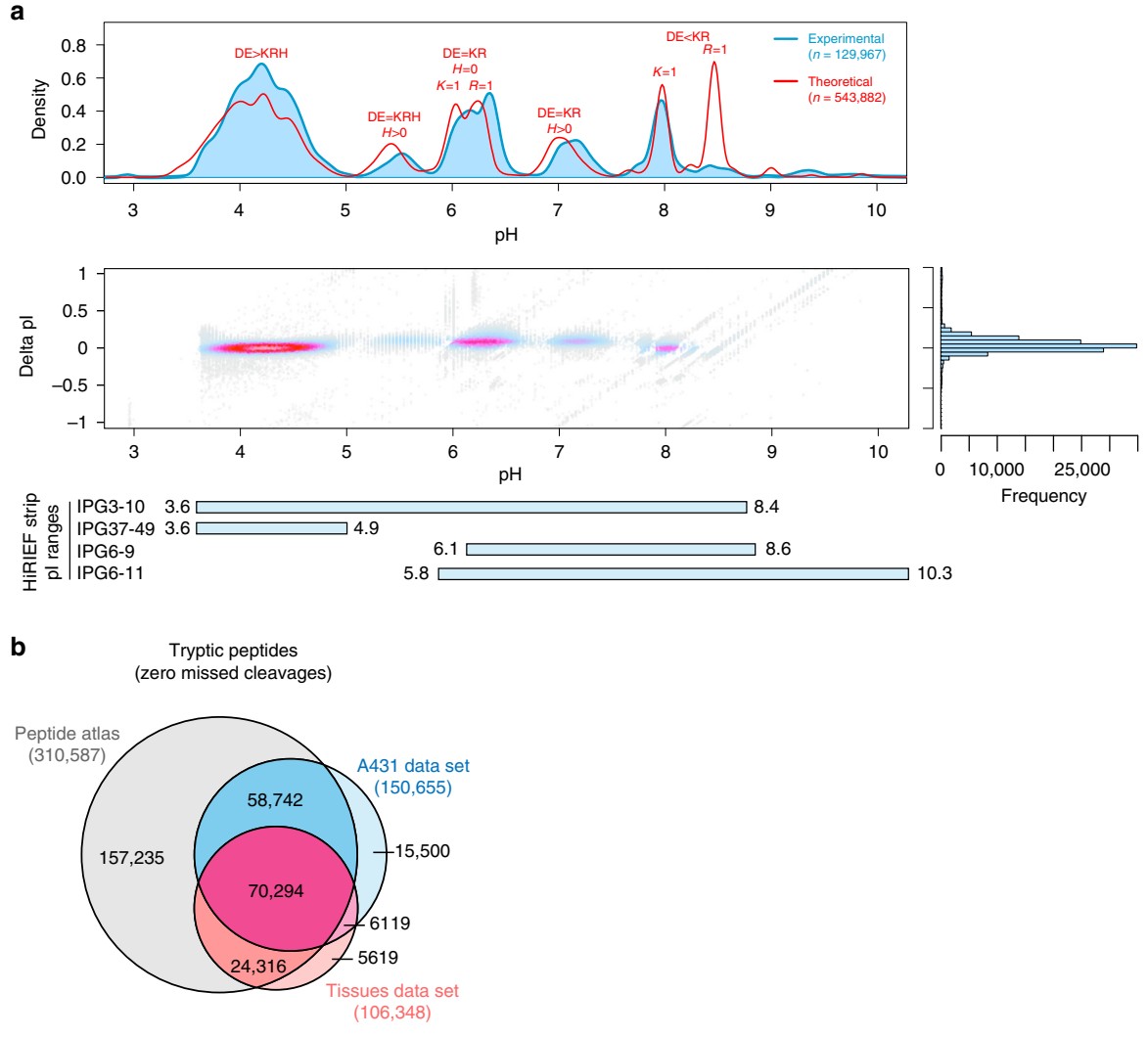

**Fig. 1** Full pI range (3–10) HiRIEF provides broad peptidome and proteome coverage. **a** The top panel shows the comparison between experimental and theoretical pI distributions of TMT-labeled peptides from the A431 cell line data set. The six major peaks in the theoretical pI distribution represent groups of peptides with characteristic amino acid compositions. For example, peptides with a higher number of Asp (D) and Glu (E) residues than the total number of Lys (K), Arg (R), and His (H) will have a pI between 3.5 and 5. The middle panel shows the accuracy of pI prediction by the PredpI algorithm[1] across the full pI range. The bottom panel shows the experimental pI ranges of the four IPG strips employed in this study. Nominal pH ranges are indicated on the left side with actual pH ranges next to the bars. See Supplementary Figures 4, 6, and 7 for pI fraction resolution, reproducibility and yield. **b** Overlap of identified fully tryptic human peptides (at protein level FDR 1%) between the A431 cells data set, the normal tissues data set and the public peptide repository PeptideAtlas (release 2017-01)

PeptideAtlas repository (release 2017–01) and added 21,619 human peptides not present in that repository (Fig. 1b).

To benchmark our data, we compared these results to those from corresponding tissues (placenta, kidney, liver, tonsil, and testis) in the Wilhelm et al. draft proteome work[21] by re-analyzing their raw data with the same search pipeline (MSGF+, Percolator, with peptide and protein-level FDR 1%) as employed for our data set. We identified 56% more peptides (169,471 vs. 108,402) and 24% more genes with protein products (10,889 vs. 8796), and consequently, protein identifications were generally backed up by larger numbers of unique peptides (i.e., 90 vs. 75% of protein identifications supported by at least 2 peptides) (Supplementary Figure 3, which includes MS run time comparison). Additionally, we performed a simple test and verified that not a single peptide from olfactory receptor proteins could be found in either of our data sets[22].

The obtained experimental peptide pI data permitted the upgrade of the PredpI algorithm[1] with updated pK constants

(available in Supplementary Data 1) now valid for the full pI range. Applying PredpI prediction to peptide sequences from an in silico tryptic digestion of the Ensembl75 human protein database enabled the alignment of the theoretical tryptic peptide distribution over the full pI range with the experimental one (Fig. 1a). The low experimental yield of peptides with pI ~8.5 could perhaps be explained by polyacrylamide gel instability in the most alkaline range[23]. For the experimental distribution, only tightly focused peptides (i.e., present in only one or two consecutive fractions, on average 80% of all peptides in the used IPG ranges, Supplementary Figure 4) were considered. This tryptic-peptide pI distribution appears to be ubiquitous to all organisms (Supplementary Figure 5) and based on the acid-base chemistry of amino acid residues and the similar distribution of amino acid frequencies in all species[24]. Peptide fractionation was reproducible across samples and IPG ranges (Supplementary Figure 6) and the unique peptide yield of pI fractions was characteristic to each IPG strip used (Supplementary Figure 7).

**Table 1 Identification statistics from the standard proteome search**

| | PSMs | Unique peptides | Proteins | Gene symbols |
|---|---|---|---|---|
| | (1% FDR) | (1% FDR) | (1% FDR) | (1% FDR) |
| *A431 data set* | | | | |
| IPG3-10 | 367,570 | 141,071 | 9816 | 9425 |
| IPG37-49 | 314,305 | 93,329 | 9679 | 9236 |
| IPG6-9 | 318,446 | 72,511 | 8584 | 8561 |
| IPG6-11 | 263,653 | 76,326 | 9056 | 9134 |
| Total | 1,263,974 | 203,640 | 11,171 | 10,166 |
| *Normal tissues data set* | | | | |
| set1 IPG3-10 | 374,966 | 118,929 | 10,096 | 9621 |
| set1 IPG37-49 | 228,675 | 63,990 | 9025 | 8817 |
| set1 total | 603,641 | 137,134 | 10,339 | 9869 |
| set2 IPG3-10 | 476,966 | 110,783 | 9660 | 9289 |
| set2 IPG37-49 | 188,988 | 74,981 | 9903 | 9700 |
| set2 total | 665,954 | 135,987 | 10,519 | 10,167 |
| Total | 1,269,595 | 169,471 | 11,471 | 10,889 |

Numbers of gene symbols (with protein products), proteins, peptides, and PSMs identified from all the IPG-strip ranges employed on A431 cells and normal tissues (placenta, liver, kidney, tonsil, liver, and testis) by searching the human Ensembl75 proteome database. See Supplementary Fig. 1 for sample layout in the two data sets and overlap in identification between IPG ranges

**A three-stage proteogenomics workflow.** To tackle the high rate of false discoveries in proteogenomics findings while taking into account the recent recommendations in the field[3, 19], we developed IPAW, a proteogenomics workflow to identify and curate novel peptides from undiscovered protein-coding loci as well as variant peptides coded by nsSNPs and mutations (Fig. 2). The workflow contains three major stages: discovery, curation, and validation. To illustrate the discovery stage, we carried out two types of proteogenomics searches, the VarDB concatenated database search (a customized database search strategy applicable for any high-resolution LC-MS/MS data) (type 1 search in Fig. 2) and the 6FT search, which uses peptide pI values to restrict database size (applicable if pI-based fractionation is used)[1, 25] (type 2 search in Fig. 2). It is also possible to use any customized databases generated by external tools[14–17]. All MS/MS spectra were searched by MS-GF+[26], and post processed with Percolator[27] under the Galaxy platform using a separate target-decoy strategy. The human VarDB consists of peptides originating from previously annotated nsSNPs, somatic mutations, pseudogenes, and long non-coding RNAs, forming a supplementary set of peptides to the canonical proteome. Customized database searches have the advantage over 6FT searches in that they allow detection of splice-junction-spanning novel peptides and have a relatively smaller search space, while the 6FT searches allow unbiased screening of protein-coding sequences in the whole genome. The discovery stage outputs a list of candidate novel and SAAV peptides with 1% class-specific FDR[3].

At the curation stage, some of the candidate novel peptides are removed. First, peptides are processed through BLASTP[28] to remove exact matches to the known protein database, which combined known human proteins from the Uniprot reference proteome (UP000005640), Ensembl 83, Refseq, and Gencode v24. Thereafter, the subset of novel and SAAV peptides with single amino acid changes compared to known peptides are curated by a tool, SpectrumAI, which only keeps those peptides whose MS2 spectrum contains product ions directly flanking both sides of the substituted residue. We then use BLAT[29] to remove peptides mapping to multiple locations in the genome. Even though these

peptides are as experimentally valid as other peptides, we decided not to prioritize them because of the increased difficulty in assigning orthogonal support and in drawing biological conclusions from proteins with uncertain genomic locations.

In the validation stage, curated novel peptides are assessed by distribution plots of intrinsic properties such as delta pI (pI difference between experimental and predicted pI values), precursor mass error, and match score. Finally, the curated novel peptides are examined in the context of their genomic positions by cross validation against multiple levels of orthogonal data. These levels include: (i) RNA-seq acquired from the same samples, (ii) public domain Ribo-seq for support of alternative translation initiation sites (TIS)[30, 31], (iii) public domain CAGE data for evidence of transcription start sites (TSS)[32], (iv) vertebrate conservation (PhyloP)[33], (v) codon substitution frequency (PhyloCSF)[34], which predicts coding potential of DNA sequences based on codon composition and substitution frequency by comparative genomic analysis, and (vi) MS data from the two draft proteome publications[21, 35]. It should be noted that the validation stage does not use any of these orthogonal data as filtering criteria to discard proteogenomic findings, but only provides a means for users to rank candidate novel peptides based on orthogonal support.

**SpectrumAI verifies single amino acid substitutions.** Novel and SAAV peptides were output by 1% class-specific FDR[3], which is essential but insufficient to guarantee a de facto 1% false discovery rate. In practice, unexpected isobaric modifications or amino acids reshuffles in known peptides could be mistakenly identified as single-substitution peptides, since the identification of peptide sequence is a pattern matching process, where the search engine only guarantees the best sequence match available in the database. Due to this uncertainty, novel peptides with single substitution were discarded in a recent proteogenomics study[20]. Nonetheless, manual inspection of MS/MS spectra should allow for distinction between correct and incorrect assignments. However, this is a laborious process and therefore inapplicable to large data sets. A previously published tool, CAMV[36], allows computer-aided manual inspection on peptide spectra matches. Although it reduces workload of manual validation, it still needs a mass spectrometry expert to make the final call to accept or reject. CAMV is more suitable to validate a limited list of peptides. For peptides with single amino acid changes, we developed a tool, SpectrumAI, which automatically eliminates incorrect peptides by inspecting MS/MS spectra for flanking product ions supporting the amino acid substitution. We demonstrated the ability of SpectrumAI to eliminate incorrect single substitution peptide identifications via (i) analysis of the distribution of precursor mass error, (ii) orthogonal evidence at DNA and RNA level, and (iii) validation using the corresponding synthetic peptides.

In the A431 data set, 4492 SAAV peptides were identified at 1% class-specific FDR. Of these, 1332 have MS/MS ions flanking the substitution and thus passed SpectrumAI curation (Supplementary Data 2). Comparing the distribution of precursor mass errors, curated peptides showed a distribution similar to that of identified known peptides, whereas discarded peptides did not (Fig. 3a). Importantly, our observation that search engine score (SpecEvalue) distributions of curated peptides and discarded peptides were similar and indicates the need for independent evidence in addition to strict FDR control, which itself depends primarily on search engine score (Supplementary Figure 8). Further, we validated the SAAV peptides using genomics and transcriptomics data. About 44% of peptides passing SpectrumAI curation were supported either at RNA or DNA level, whereas only 3% of discarded peptides had such support (Fig. 3b). We

compared the frequency of amino acid changes of the 735 curated SAAVs lacking orthogonal support with that of the 426 supported both at DNA and RNA level, and observed that certain types of amino acid changes (the top one being glutamine to glutamic acid) were over-represented (Supplementary Figure 9). A plausible explanation could be that some curated SAAV peptides are chemical artifacts, arising by, e.g., deamidation.

Finally, we selected 30 peptides with single amino acid substitution from the discovery stage (of which 19 passed SpectrumAI curation, whereas 11 did not), and purchased the corresponding synthetic peptides for validation. Manual inspection of the mirror plots (Fig. 3c and Supplementary Data 3, pp. 1–30) confirmed the 19 SpectrumAI curated peptides to be correct, and the 11 discarded ones to be incorrect.

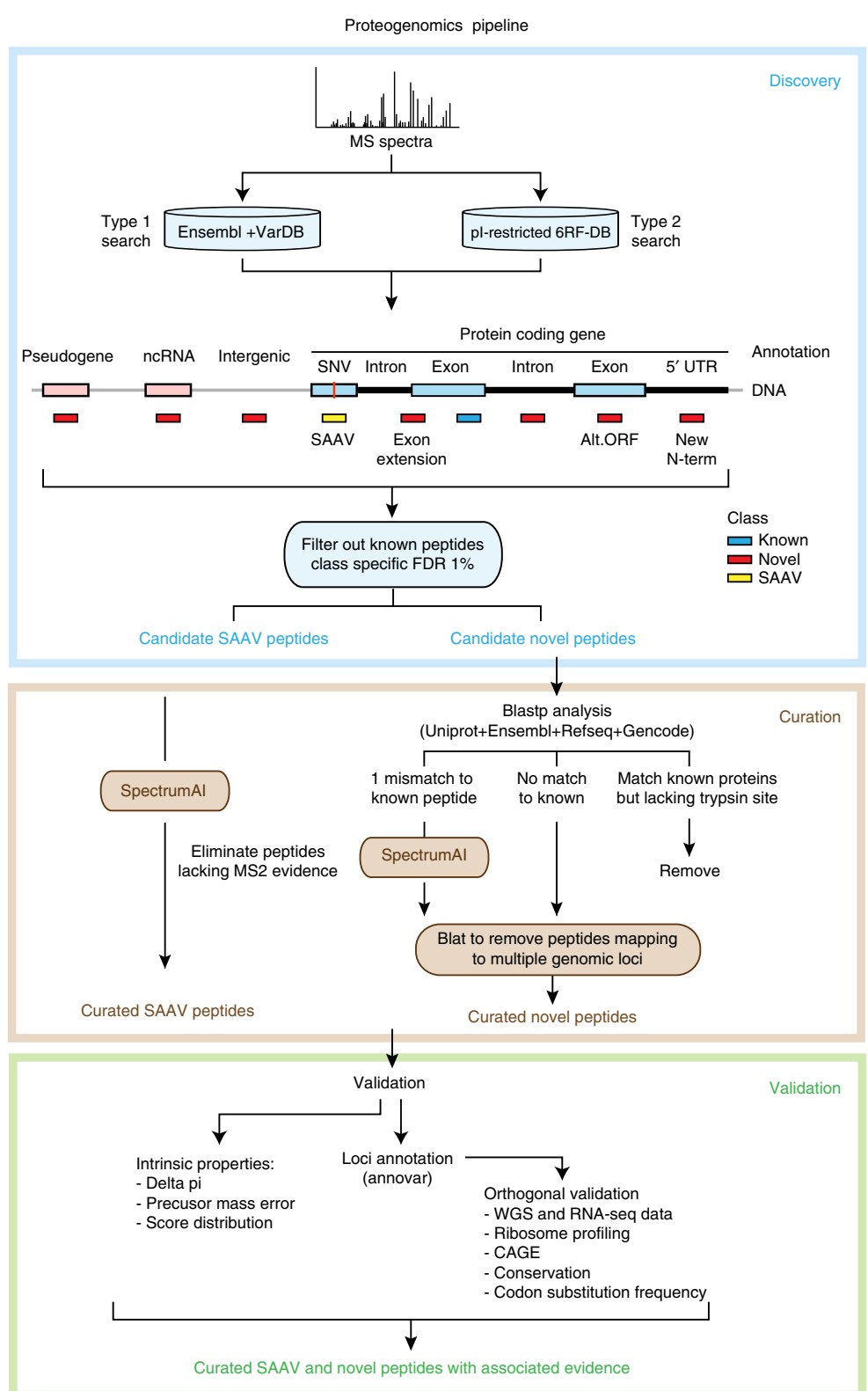

**Novel peptides from unannotated protein-coding loci.** IPAW was applied to discover unannotated protein-coding loci in the human genome using both the A431 cells data set and the normal tissues data set. The combined result of 6FT and VarDB searches (Supplementary Figure 10), at the discovery stage, yielded 710 and 295 novel peptides (class-specific FDR 1%) in A431 cells and normal tissues, respectively, of which 426 and 155 passed the curation stage (Supplementary Figure 11 and Supplementary Data 4 and 5). Precursor mass error, SpecEvalue and delta pI distributions of these curated peptides were assessed in comparison to those of known canonical peptides (Supplementary Figures 12–14). Novel peptides within genomic proximity of 10 kb were grouped and considered to belong to the same locus. Thus, we identified 374 and 140 unknown protein-coding genomic loci in A431 cells and normal tissues, respectively, of which 42 and 13 were supported by two or more peptides (Fig. 4a and Supplementary Figure 15a). Novel peptides were categorized into eight different classes: pseudogene, 5′ un-translated region (UTR), intronic, AltORF, ncRNA, exon extension, intergenic, and 3′ UTR (Fig. 4b and Supplementary Figure 15b) and were found in all chromosomes across the genome (Fig. 4c and Supplementary Figure 15c).

The largest group of novel peptides identified belonged to pseudogenes, which, given the previous observation of thousands of pseudogenes found expressed at RNA level in cancer[37], is maybe to be expected, particularly in cancer cell lines such as A431. There were 30 novel peptides identified from sORFs (the majority being AltORFs) in known coding transcripts after examination of the genomic context. A mass spectrometry study performed by Slavoff et al. identified 90 sORF-encoded peptides in human cells[11]. Four peptides were common between our list and their study despite the fact that they used a different cell line (human leukemia cell line K562) and workflow. In our present study, regarding protein N-terminal extensions, 45 out of 50 cases used non-AUG near-cognate translation start codons, the majority (30 cases) of which was within a strong Kozak motif (Supplementary Data 4). A non-AUG TIS within a strong (or occasionally moderate) but not optimal Kozak sequence may be characteristic of protein N-terminal extensions. The usage of non-AUG start codons is corroborated by the work of Fritsch et al., a ribosomal footprinting-based study, which reported that only 1% of protein N-terminal extensions used canonical AUG start codons[30]. Novel peptides belonging to AltORFs and protein N-terminal extensions showed stronger support from orthogonal data compared to novel peptides from other groups (Fig. 4d and Supplementary Figure 15d). Although most peptides mapping to pseudogenes in the current study lacked RNA-seq read coverage, external evidence from ribosome profiling and CAGE was more supportive. Moreover, conservation scores of pseudogenes/non-coding RNAs with novel peptides identified were significantly higher than those of 1000 randomly selected pseudogenes/non-coding RNAs (Fig. 4e).

Since signal peptides are usually located in the N termini of proteins, we tested the hypothesis of whether an N-terminal extension could alter protein subcellular localization[38] by using the TargetP algorithm (v1.1)[39] to predict subcellular localization. Three of the discovered protein N-terminal extensions, in genes CDK16 (identified by 2 peptides), NPLOC4 (2 peptides), and THOP1 (2 peptides), showed a high likelihood of being targeted to the mitochondria while the respective canonical proteins have cytosolic or nuclear location (Supplementary Data 6). This result suggests that, for certain genes, the existence of an upstream TIS in the same reading frame as the canonical AUG start site may be used to submit the subcellular location of the protein products to control at the translation level.

Other types of events found include protein products of exon extensions and intron retentions. Interesting examples of the latter are the translations of alternatively spliced (or miss-spliced) mRNAs from EGFR (identified by 2 peptides) and AP1S1 (2 peptides). The resulting polypeptide products contain part of the canonical N terminus followed by a completely new intron-derived C-terminal sequence, 181 and 42 amino acids long for EGFR and AP1S1, respectively. It is unclear whether such observations are errors of splicing machinery or a functional cellular response to stress conditions.

Finally, to further validate the novel peptides, 100 synthetic peptides in addition to the 30 mentioned above were purchased and analyzed. Out of these 100 synthetic peptides, 2 failed to generate good fragmentation spectra, 7 demonstrated their endogenous counterparts to be incorrect identifications upon manual inspection, and the remaining 91 novel peptides were successfully validated (Supplementary Data 3, pp. 31–128). The significant number (7 out of 98) of incorrect novel peptides highlights the need for further efforts into curation/validation of findings in the proteogenomics field.

For submission to UniProt, 40 unannotated coding loci with multiple peptides that could be linked to in-frame initiation start codons were included (Supplementary Data 7, with selected examples showing each event type in Fig. 5).

**TMT-based quantitative proteogenomics analysis.** Tissue specificity or regulation upon stimuli can hint at functional roles of the discovered proteins. Some pseudogene proteins (e.g., those with GAPDH gene ancestry) showed broad tissue expression similarly to that of their parental gene (Supplementary Data 8). Others showed protein expression prevalent to particular tissues. For example, pseudogene proteins TATDN2P1 (identified and quantified by 2 peptides, 3 PSMs) and UBE2L5P (2 peptides, 6 PSMs) were specific to testis, whereas their parental gene proteins UBE2L3 and TATDN2 were broadly expressed on all measured tissues (Fig. 6 and Supplementary Data 8), suggesting independent functions for these two "pseudogenes" in testis. RNA-seq reads for both these pseudogenes support their testis specificity (Fig. 6d and Supplementary Data 5). Additionally, several novel peptides with placenta specificity were found from two lncRNAs, i.e., TINCR (also known as PLAC2 - placenta specific 2, which was identified by 1 peptide, 1 PSM) and CTD-2620I22.3 (ENSG00000267943, 4 peptides, 5 PSMs) (Figs. 5b and 6a, and Supplementary Data 8), both also showing placenta specificity at transcript level (Supplementary Figure 16). TINCR is thought to regulate differentiation in epidermal tissue, and our present result

**Fig. 2** A proteogenomics workflow to discover, curate, and validate novel and SAAV peptides. The pipeline consists of three major stages: discovery, curation, and validation. The discovery stage is performed with MS-GF+ using two database strategies. Type 1 search was performed against a single database consisting of known peptides concatenated with variant peptides. Type 2 search is enabled by HiRIEF peptide fractionation and was performed against pI-restricted databases of tryptic peptides generated from a six-frame translation (6FT) of the human genome. The discovery stage outputs 1% class-specific FDR for novel and SAAV peptides. In the curation stage, candidate SAAV peptides are curated by SpectrumAI. The novelty of candidate novel peptides from the discovery stage is ensured by BLASTP analysis against known protein databases including Uniprot reference proteome (with isoforms), Ensembl human protein database v83, RefSeq and Gencode v24, and the subset of novel peptides with single amino acid substitution are also curated by SpectrumAI. In the validation stage, quality control plots such as delta pI, precursor mass error, and search engine score distribution are made. In addition, curated novel peptides are evaluated for orthogonal data support in, e.g., RNA-seq data, ribosome profiling and CAGE data, conservation and coding potential prediction

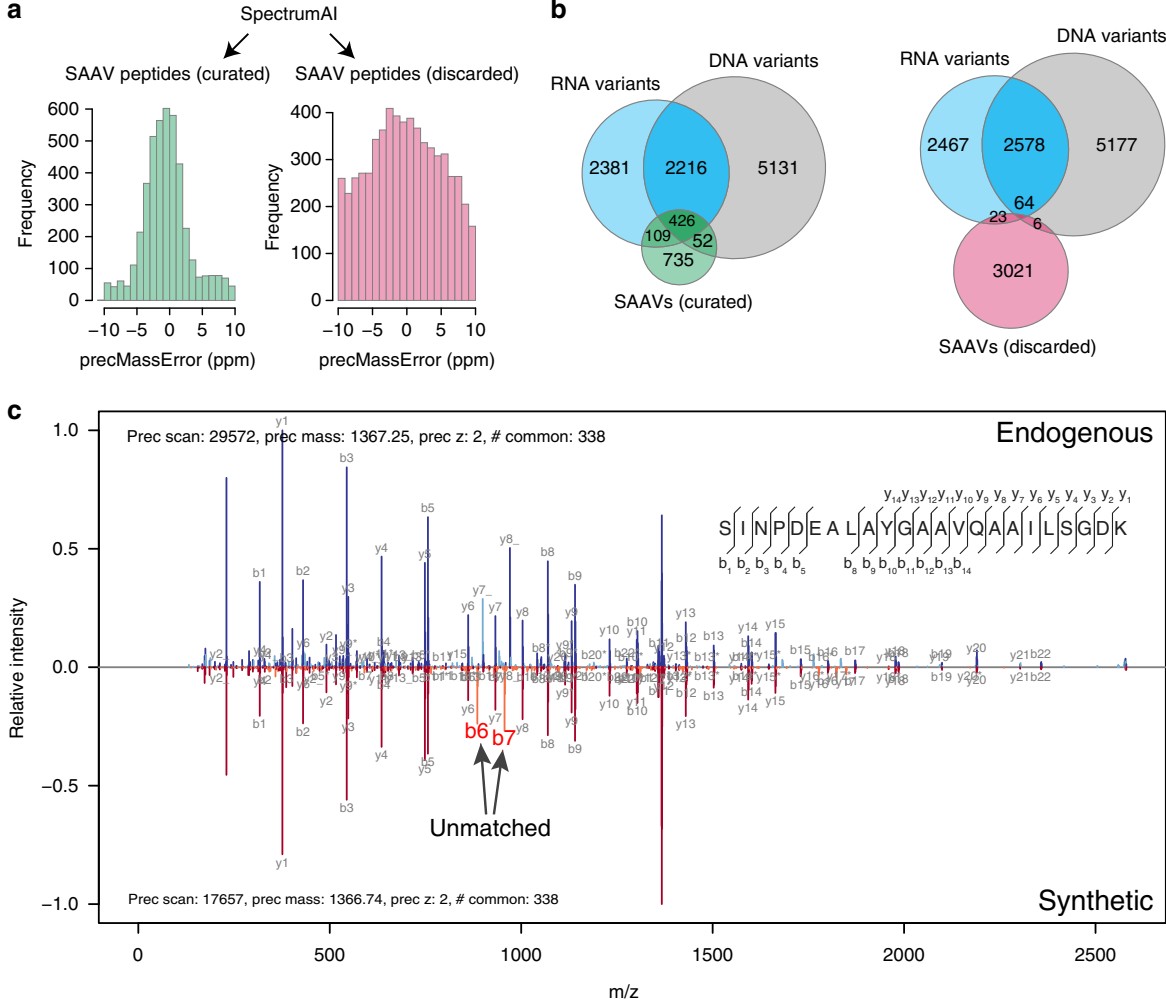

**Fig. 3** SpectrumAI increases identification accuracy of peptides with single amino acid changes. **a** Precursor mass error distributions of peptides classified as curated and discarded by SpectrumAI. **b** Curated SAAV peptides have more overlap with missense variants identified at DNA and RNA level. **c** Mirror plot of an incorrectly identified peptide (that yet had passed discovery stage with class-specific FDR 1%) with a single residue substitution (V > L, at position 8) that was subsequently discarded by SpectrumAI. Annotated MS2 spectrum of the endogenous peptide is shown on top, whereas that of the respective synthetic peptide is inverted and shown on bottom. This incorrect peptide identification detected by SpectrumAI shows mismatching b6 and b7 product ions (highlighted in the synthetic side and missing in the endogenous side), which ought to have flanked the substituted residue, indicating that the endogenous amino acid sequence is incorrect between its sixth and eighth residues

suggests that such molecular function is not restricted to the RNA product of the gene but may extend also to the protein product.

Upon *EGFR* inhibition of A431 cells, some of the discovered pseudogene proteins, e.g., *RBBP4P1* (identified and quantified with 3 peptides, 8 PSMs), followed the same quantitative pattern as the parent gene proteins (Supplementary Data 8). There were also examples of pseudogenes, such as *HSPA8P1* (3 peptides, 10 PSMs) and *ANAPC1P1* (1 peptide, 2 PSMs, out of only two unique theoretical tryptic peptides differing from the canonical parent *ANAPC1*), showing very distinct quantitative patterns from their parent genes, suggesting that, in these latter cases, the pseudogenes possess an independent transcriptional control. We investigated whether, in general, protein-level regulation between pseudogenes and the respective parent genes correlated. Out of 242 pseudogene-parent gene pairs (median correlation coefficient 0.26), 20 correlated significantly (test of association, *p* value <0.01), 16 were positively correlated (mean Pearson correlation 0.88), and 4 were negatively correlated (mean Pearson correlation −0.79) (Supplementary Figure 17).

Some AltORF proteins such as Alt-*HNRNPUL2* (identified and quantified with 3 peptides, 7 PSMs) and Alt-*DRAP1* (4 peptides, 14 PSMs), resulting from upstream translation were found in the A431 data set to be downregulated 24 h after treatment, whereas their canonical counterparts showed slight upregulation (*HNRNPUL2*) or no clear regulation (*DRAP1*) (Fig. 5c and Supplementary Data 8). Similarly, N-terminal extended proteins resulting from in-frame upstream TIS such as those from *RAB12* (2 peptides, 2 PSMs) and *WDR26* (1 peptide, 4 PSMs, from a UTR region which accommodates only one theoretical tryptic peptide of MS-amenable size) showed downregulation after treatment while their respective canonical proteins remained constant. Translational competition between upstream TIS and canonical TISs has been observed previously, in accordance to the "leaky scanning" model of translation[38]. Here, we observed four cases of usage of upstream TIS decreased by the *EGFR* inhibition treatment.

## Discussion

The majority of proteomics studies still neglects sequence variants and proteins arising from unannotated coding regions because only reference proteome databases are searched. The emerging field of proteogenomics allows the identification of previously

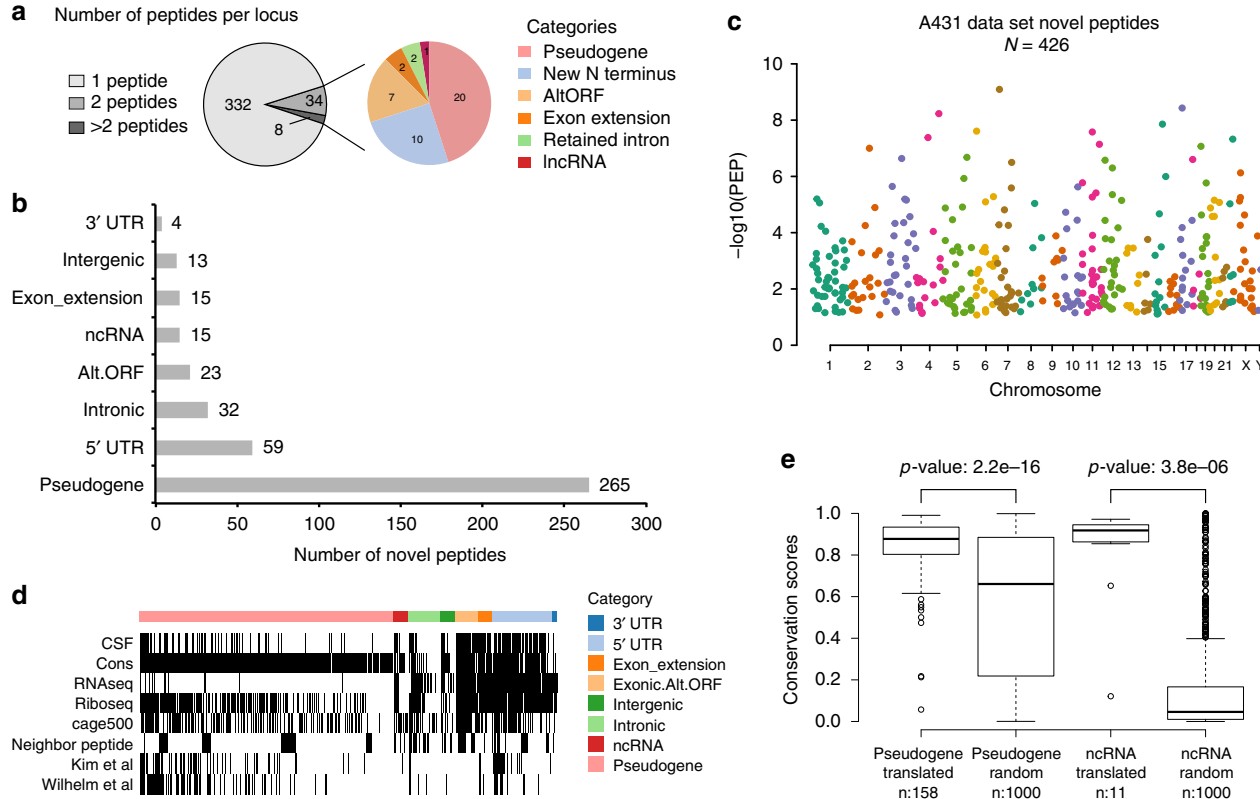

**Fig. 4** Unannotated protein-coding loci found in the A431 cells data set. **a** The left pie chart shows the number of unannotated protein-coding loci supported by one, two, or more peptides (peptides within 10 kb distance were grouped into one locus); the right pie chart shows the different types of unannotated coding events supported by multiple peptides. **b** Automatic categorization of novel peptides by annovar using RefSeq gene annotation (see "Methods"). **c** Manhattan plot of novel peptides, where the y-axis represents the peptide's posterior error probability (PEP). **d** Orthogonal data support for novel peptides, including PhyloCSF coding potential, conservation analysis, A431 cell line RNA-seq reads evidence, ribosome profiling, CAGE (up to 500 bp upstream from peptide location), presence of neighboring peptides (within 10 kb), and whether the peptide was identified in the draft proteome data of Kim et al.[35] and Wilhelm et al.[21]. Continuous variables were discretized to binary values 0 or 1 for visualization purposes. 10,000 random genomic loci were used to determine the threshold to call if Ribo-seq or CAGE data were supportive or not (see Supplementary Figure 20). **e** The conservation score (PhastCons[68] score) distribution of pseudogenes and lncRNAs for which peptides were found was compared to that of 1000 randomly selected pseudogenes and lncRNAs. In the box plots, center line corresponds to median, box boundaries correspond to the first and third quartiles (Q1 and Q3), the upper whisker is min(max(x), Q3 + 1.5 × IQR) and lower whisker is max(min(x), Q1–1.5 × IQR)

unknown proteins and protein variants, a process which should ultimately lead to complete genome annotation and improved understanding of proteome biology. In this context, we extended the HiRIEF method to enable detection of peptides across the full pI range of the tryptic peptidome, eliminating the previous bias to acidic peptides. Moreover, a proteogenomics workflow, IPAW, was developed to make use of the improved HiRIEF MS data in the 6FT search strategy. IPAW is also applicable to widely used concatenated database search strategies and focuses on curation and validation of putative novel peptides using multiple independent lines of evidence to achieve higher sensitivity and reliability.

One important application of proteogenomics is to identify protein variants derived from cancer-specific mutations[40–42]. Despite recent efforts, this area has been troublesome due to a high risk of false-positive identifications. Class-specific FDR[3] based on search engine scores is required to control error rates in novel peptide findings but, as we show here, it is by itself insufficient because some high-scoring identifications with apparently good matching fragmentation spectra actually turned out to be incorrect (Fig. 3c and Supplementary Data 3). Addressing this problem, SpectrumAI was shown here to be capable of efficiently eliminating incorrectly identified single substituted peptides with high accuracy. Compared to a previous computer-aided spectrum inspection program, CAMV[36], SpectrumAI was specifically

designed to curate single substitution peptides with high accuracy without the need for manual inspection. Although it is possible to reduce the risk of false positives by including in the search space only variant peptides from nsSNPs or mutations identified by genomics or transcriptomics, incorrectly called SNPs or mutations from sequencing data could propagate into false peptide identifications and, conversely, false negatives due to low RNAseq read counts are also possible. The problem could worsen when generating a database by combining a large list of non-synonymous variants from multiple sequenced samples. Additionally, SpectrumAI can be used to curate proteomics data and provide false-positive control on single substitution peptide identifications for studies that lack corresponding sequencing data. SpectrumAI is limited to validate peptides with single amino acid changes, and not designed for peptides with inserted or deleted amino acids. With that said, we believe it will become particularly useful in proteogenomics studies aiming to develop immunotherapy against cancer in which identification of cancer-specific mutations at protein level is crucial to discover potential neoantigens[43].

Further, we here provide the proteogenomics tools (see "Data and Software availability" in the "Methods" section) to assist users in validation of findings by orthogonal data. Four major types of data (RNA-seq, conservation, Ribo-seq, and CAGE) can be useful

to prioritize the mass spectrometry findings. These data should be used with caution and considered under appropriate context. For instance, Ribo-seq data are particularly useful to validate discovered upstream translation initiation events, but it is not conclusive

translation evidence for non-coding transcripts or intergenic sequences. Conservation between species is useful to evaluate novel peptides found in intergenic regions, but not for peptides overlapping with the CDS region. CAGE data only indicates

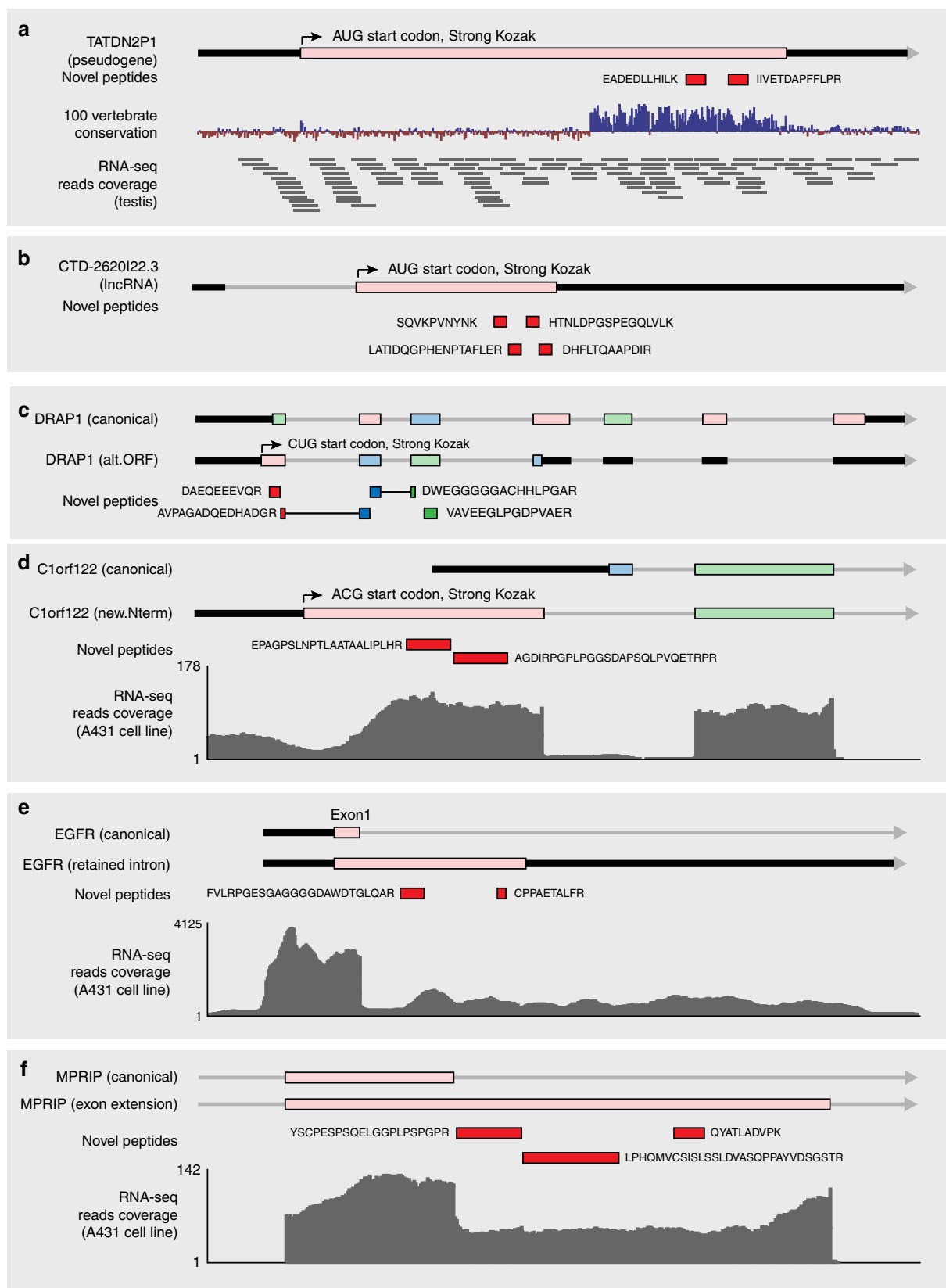

transcription start sites. Finally, RNA-seq data can provide additional transcriptional expression evidence for translation of intergenic or intronic sequences, but not for protein N-terminal extensions or AltORF proteins overlapping with known coding regions.

IPAW enabled confident discovery of proteins from supposedly non-coding genomic regions such as ncRNAs (pseudogenes and lncRNAs) and UTRs. Recently, a bioinformatic study which re-analyzed ribosome profiling data indicated that ~40% of lncRNAs and pseudogenes are translated, and that ~35% of mRNAs are translated upstream of the primary protein-coding region (uORFs)[44]. Ribosomal binding is indicative of translation initiation but not ultimate evidence of encoded peptides. In fact, Guttman et al. stated that ribosomal occupancy is not necessarily equivalent to translation and that the majority of lncRNAs do not encode proteins[45]. Similarly, Banfai et al. analyzed MS and transcriptomics data in two human cell lines and showed that long non-coding RNAs were rarely translated[46]. This may explain why we identified relatively few peptides encoded from lncRNAs. Some pseudogenes have been shown to play an active role in cancer development and progression, though in most cases, only RNA transcripts have been considered active molecules[37, 47, 48]. Evidence pointing to translation of pseudogenes (Figs. 4 and 6) suggests that, at least for some of them, molecular function could actually be carried out by their protein products, which in fact has been previously shown for the *NANOGP8* pseudogene[49]. It was somewhat surprising that a significantly higher number of pseudogene peptides was found in A431 cells than in normal tissues (266 vs. 104 peptides, respectively), and that a similar situation occurs for AltORF and N-terminal extension peptides (29 vs. 0 and 56 vs. 13 peptides, respectively) (Supplementary Data 4 and 5), all the while considering that the analytical depth of the two data sets was similar (Fig. 1b). This may suggest that cancer cells are more prone to produce these unusual proteins, but further studies are needed to ascertain this hypothesis.

There is emerging evidence for the translation of short open reading frames (sORFs)[4], many of them upstream ORFs. Fritsch et al. used ribosomal footprinting to probe the genome for undiscovered uORFs and N-terminal protein extensions, and identified ~3000 previously unannotated uORFs, 546 N-terminal protein extensions, and more than 1000 AltORFs[30]. In our present study, 56 peptides were identified from N-terminal protein extensions and 29 peptides from AltORF proteins in A431 cells (Supplementary Data 4). A common observation between these proteomics studies and ribosome profiling studies is that N-terminal protein extensions and AltORF proteins are mostly originating from non-AUG near-cognate start codons. Moreover, our results indicate that most of them are within a strong Kozak sequence, suggesting that a near-cognate translation initiation site (TIS) within a strong Kozak sequence may be prerequisite for uORF translation. Observation of these events by MS is less frequent than by ribosome profiling. The reason for this could be that either the ribosome skips over these non-AUG near-cognate start sites or the translation does occur but with low efficiency, leading to low abundant polypeptides difficult to detect by MS. It should be noted that, since AltORF proteins and N-terminally extended proteins arise from the same transcript as their canonical

counterparts, transcriptomics technologies are blind to both, and therefore these events can only be discovered by proteogenomics.

"Retained intron" proteins were yet another type of interesting findings. These result from alternative splicing of pre-mRNAs, possibly even splicing abnormalities due to some form of spliceosome impairment. The resulting polypeptide typically kept the native protein's N-terminal segment and combined it with a completely new C terminus derived from the intron. How "retained intron" proteins such as the ones found here for the *EGFR* and *AP1S1* genes (Fig. 5e and Supplementary Data 7, pp. 18) escape mRNA surveillance mechanisms such as nonsense-mediated mRNA decay (NMD) remains to be investigated. Nonetheless, because of their intron-derived C terminus with rather unusual amino acid sequence, these proteins, if proved to be cancer specific events, could perhaps elicit stronger immunogenic response than neoantigens from single amino acid changes. In this respect, proteins from retained introns are theoretically on par with AltORF proteins derived from non-canonical reading frame translations[50–52].

In summary, we have developed IPAW, a comprehensive proteogenomics workflow able to curate and validate various types of novel peptides, and we have applied it to two in-depth proteomics data sets acquired from an extended HiRIEF method. The curation and validation workflow is compatible with any search strategies such as pI-restricted 6FT searches, or concatenated database searches using VarDB or databases derived from sequencing data. Moreover, SpectrumAI is provided to curate single substitution peptides by automatic examination of MS2 spectra. And further, with a focus on validation, we provide a set of tools for users to search orthogonal support of putative novel peptides in conservation data, RNA-seq data, CAGE, or ribosome profiling data. The here described methods should assist the proteogenomics field to reliably integrate protein-level information into genome annotation and furthermore should be particularly useful in understanding the cancer proteome and may lead to the discovery of therapeutic targets against cancer.

## Methods

**Uppsala University normal tissue samples ethical permits**. Human tissue samples were collected under individual informed consent and handled in accordance with Swedish laws, and obtained from the Department of Pathology, Uppsala University Hospital, Uppsala, Sweden, as part of the sample collection governed by the Uppsala Biobank (http://www.uppsalabiobank.uu.se/en/). All human tissue samples were anonymized and used in accordance with approval and advisory report from the Uppsala Ethical Review Board (Reference #2002-577, 2005-338 and 2007-159 (protein) and #2011-473 (RNA)).

**Cell lysis, protein extraction, digestion, and TMT labeling**. The A431 cells (DSMZ #ACC-91, tested for, and found free of, Mycoplasma contamination) and normal tissues were lysed in SDS-lysis buffer (4% (w/v) SDS, 25 mM HEPES pH 7.6, 1 mM DTT). Lysates were then heated at 95 °C for 5 min in a thermomixer, and were sonicated with a probe sonicator (Bandelin Sonopuls, Buch and Holm) twice using 50% duty cycle, 50% power for 15 s, in order to shear DNA. Samples were centrifuged at $14,000 \times g$ to remove cell debris, the supernatant was collected and protein concentration estimated by the DC-protein assay (BioRad) following the manufacturer's instructions. From each sample, 250 µg of total protein were taken and processed according to the FASP (Filter aided sample preparation)[53] protocol with slight modifications. Briefly, the filter units (Nanosep Centrifugal Devices with Omega Membrane, 10 K, blue, P/N OD010C34, from Pall Corporation) were first washed with 200 µl of Milli-Q water (Millipore Corporation) by centrifugation at $14,000 \times g$ for 15 min, after which the sample, previously diluted in 200 µl of urea buffer (8 M urea, 1 mM dithiothreitol, in 25 mM HEPES pH 7.6),

**Fig. 5** Examples of unannotated protein-coding regions discovered. Gray lines indicate introns, black thick lines are UTRs, colored boxes are coding regions (color indicates reading frame). Novel peptides are shown as red boxes unless they are in different reading frames. **a** Pseudogene *TATDN2P1* protein identified with two novel peptides linked in the same open reading frame. **b** LncRNA ENSG00000267943 protein identified with four novel peptides. **c** An alternative reading frame protein of the *DRAP1* gene was identified with four novel peptides. The color of exons and novel peptides indicates reading frame. Exons and peptides in same colors (darker shade for peptides) are in the same reading frame. **d** Alternative protein N terminus for gene *C1orf122* was identified with two novel peptides. **e** Two novel peptides serving as evidence for the existence of "retained intron" translation for the *EGFR* gene. **f** Extended exon protein variant of gene *MPRIP* was identified with three novel peptides

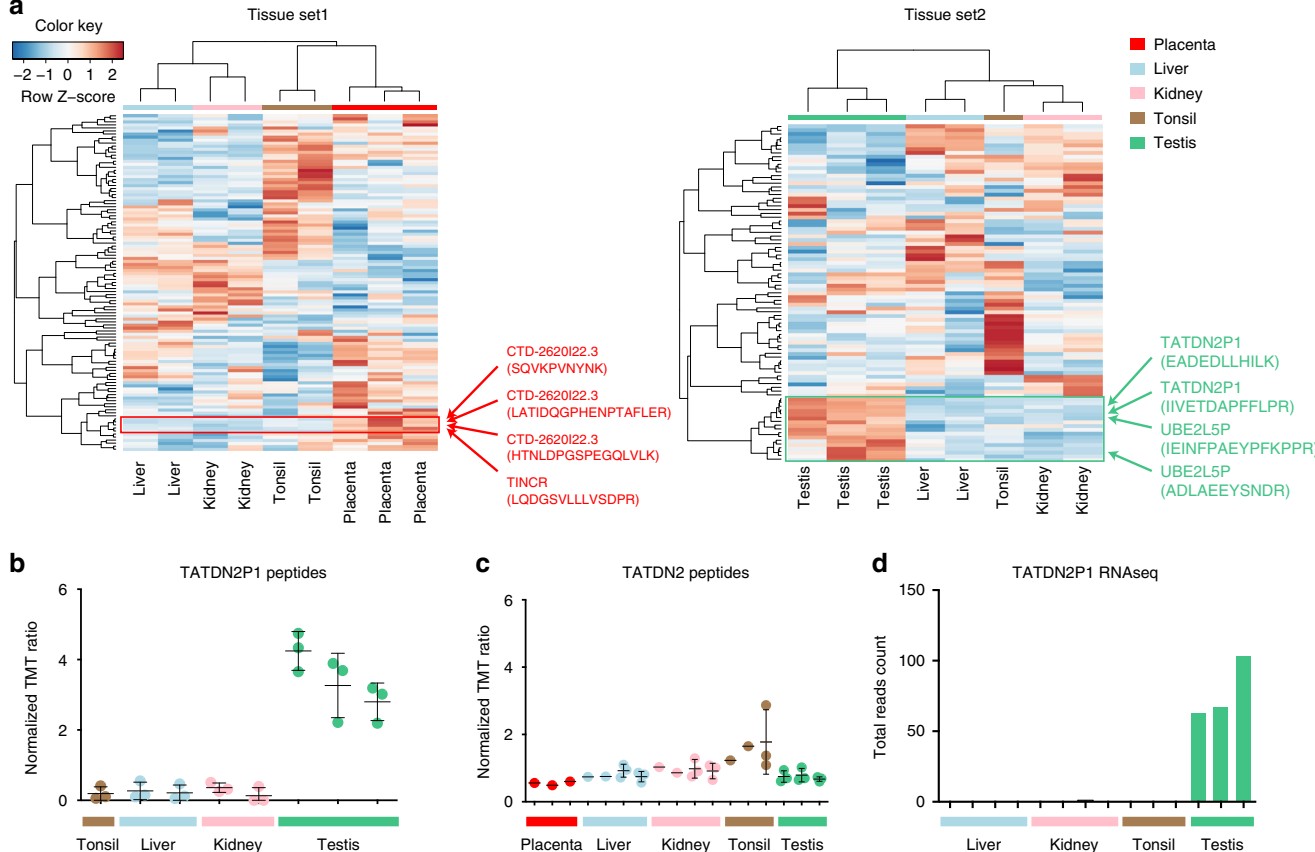

Fig. 6 Quantitative analysis of novel peptides identified in the normal tissues data set. **a** Novel peptide tissue expression. Pearson correlation and complete linkage method was used for clustering. Row Z-scores are shown in the heat map. **b** TMT-based tissue quantification of the pseudogene *TATDN2P1* peptides points to testis specificity. The three dots in the TMT ratio plots indicate quantification of three individual PSMs, with the center bar as the mean and error bars as standard deviation. **c** TMT-based tissue quantification of *TATDN2* peptides indicates broad tissue expression (quantification values from three PSMs). **d** RNA-seq read counts of *TATDN2P1* in different tissues confirms testis specificity

was loaded. After centrifugation and discarding of the flow through, there followed a wash with 200 μl of urea buffer. Urea buffer with 25 mM iodoacetamide instead of dithiothreitol was then added and the filter units were incubated with shaking for 10 min at room temperature. After centrifugation and flow-through discard, two washes were done with 200 μl of diluted urea buffer (4 M urea in 25 mM HEPES pH 7.6). After discarding the flow-through once more, 100 μl of Lys-C buffer (0.5 M urea, 50 mM HEPES, Lys-C protease from Thermo Pierce at enzyme to protein ratio 1:50 in mass) was added and the filter units were incubated with mild shaking at 37 °C for 3 h. Without centrifuging, 100 μl of trypsin buffer (50 mM HEPES, trypsin from Thermo Pierce at enzyme to protein ratio 1:50 in mass) was added on to the filter units and incubation proceeded with mild shaking at 37 °C for 16 h. Normal tissues were only digested with trypsin (not with Lys-C) overnight. The peptides were collected by centrifugation in two steps (50 μl of Milli-Q water being added before the second centrifugation) to improve yield. Peptide mixtures were transferred to fresh tubes and concentrations were estimated by the DC-protein assay (BioRad). From each sample, 100 μg of peptides were labeled with TMT10plex (Thermo Fisher Scientific) according to the manufacturer's instructions.

**High-resolution isoelectric focusing (HiRIEF) separation.** After pooling the samples that belonged together into each TMT set, each TMT set was cleaned by strong cation exchange solid phase extraction (SCX-SPE, Phenomenex Strata-X-C, P/N 8B-S029-TAK). After drying in a SpeedVac (Thermo SPD111V with refrigerated vapor trap RVT400), the equivalent to 400 μg of peptides of each sample was dissolved in 250 μl of 8 M urea, 1% pharmalyte (broad range pH 3–10, GE Healthcare, P/N 17-0456-01), and this solution was used to rehydrate the IPG dry strip (linear pH 3–10, 24 cm, GE Healthcare, P/N 17-6002-44) overnight. Focusing in IPG strips 6–9 and 6–11 was done in similar manner. For focusing in the 3.7–4.9 IPG strip, a sample bridge (with pH 3.7) was employed. In this case, 400 μg of peptides of each sample were dissolved in 150 μl of 8 M urea, and this solution was used to rehydrate the sample gel bridge overnight. The prototype narrow IPG strip 3.7–4.9 was rehydrated in 250 μl of 8 M urea, 1% pharmalyte (2.5–5 pH range, GE Healthcare, P/N 17-0451-01). Focusing was done on an Ettan IPGphor 3 system (GE Healthcare) ramping up the voltage to 500 V in 1 h, then to 2000 V in two

more hours, and finally to 8000 V in six more hours, after which voltage was held at 8000 V for additional 20 h or until 150 kVh were reached. After focusing was complete, a well former with 72 wells was applied onto each strip, and liquid-handling robotics (slightly modified Ettan Digester from GE Healthcare, which in turn is a modified Gilson liquid handler 215), using three rounds of different solvents (i. milliQ water, ii. 35% acetonitrile, and iii. 35% acetonitrile, 0.1% formic acid), added 50 μl of solvent to each well and transferred the 72 fractions into a microtiter plate (96 wells, PP, V-bottom, Greiner P/N 651201), which was then dried in a SpeedVac.

**LC-MS/MS analysis.** For each LC-MS run of a HiRIEF fraction, the auto sampler (Ultimate 3000 RSLC system, Thermo Scientific Dionex) dispensed 15 μl of mobile phase A (95% water, 5% dimethylsulfoxide (DMSO), 0.1% formic acid) into the corresponding well of the microtiter plate (96 well V-bottom, polypropylene, Greiner), mixed by aspirating/dispensing 10 μl ten times, and finally injected 7 μl into a C18 guard desalting column (Acclaim pepmap 100, 75 μm × 2 cm, nano-Viper, Thermo). After 5 min of flow at 5 μl min⁻¹ with the loading pump, the 10-port valve switched to analysis mode in which the NC pump provided a flow of 250 nL min⁻¹ through the guard column. The slightly concave curved gradient (curve 6 in the Chromeleon software) then proceeded from 3% mobile phase B (90% acetonitrile, 5% DMSO, 5% water, 0.1% formic acid) to 45% B in 50 min (65 min for normal tissues) followed by wash at 99% B and re-equilibration. Total LC-MS run time was 74 min (89 min for normal tissues). We used a nano-EASY-Spray column (pepmap RSLC, C18, 2 μm bead size, 100 Å, 75 μm internal diameter, 50 cm long, Thermo) on the nano electrospray ionization (NSI) EASY-Spray source (Thermo) at 60 °C. Online LC-MS was performed using a hybrid Q-Exactive mass spectrometer (Thermo Scientific). FTMS master scans with 70,000 resolution (and mass range 300–1700 m/z) were followed by data-dependent MS/MS (35,000 resolution) on the top 5 ions using higher energy collision dissociation (HCD) at 30% normalized collision energy. Precursors were isolated with a 2 m/z window. Automatic gain control (AGC) targets were 1e6 for MS1 and 1e5 for MS2. Maximum injection times were 100 ms for MS1 and 150 ms for MS2. The entire duty cycle lasted ~1.5 s. Dynamic exclusion was used with 60 s duration. Precursors with

unassigned charge state or charge state 1 were excluded. An underfill ratio of 1% was used.

**Standard proteomics reference database search.** All MS/MS spectra were searched by MSGF+[26], and post processed with Percolator[27] under the Galaxy platform using a separate target-decoy strategy. Tools have been used from or made available to the galaxy toolshed repository of the Galaxy-P project (https://github.com/galaxyproteomics/tools-galaxyp). The reference databases used were the human protein database of Ensembl 75. We set the precursor mass tolerance to 10 ppm; carbamidomethylation on cysteine and TMT-10plex on lysine and N terminus as fixed modifications; and oxidation of methionine as variable modification. Quantitation of TMT-10plex reporter ions was done using an integration window tolerance of 10 ppm. PSMs and peptides were filtered at 1% FDR (peptide level) and additionally proteins and gene protein products were filtered at 1% FDR (protein level) using the "picked" protein FDR method[54].

**Conversion of HiRIEF fraction numbers to experimental pI.** The IPG3–10 strip was calibrated using fluorescently labeled pI markers. Thereby IPG3–10 fraction numbers can be converted to experimental pI (exppi) values using the linear equation

$$IPG3-10 : y = 0.0676x + 3.5478 \qquad (1)$$

(where $y$ is exppi and $x$ is the IPG3-10 fraction number).

Similarly, linear equations were derived for the other HiRIEF ranges employed in this study, as follows:

$$IPG3.7-4.9 : y = 0.0174x + 3.5959 \qquad (2)$$

$$IPG6-9 : y = 0.0336x + 6.1159 \qquad (3)$$

$$IPG6-11 : y = -0.0762x + 10.3936 \qquad (4)$$

Note that IPG6–11 exceptionally has the lower fraction numbers at the alkaline end, whereas all other ranges have lower fraction numbers at the acidic end of the strip.

**PredpI algorithm updated to perform on the full pI range.** A random subset of tightly focused peptides (10,000 taken from the A431 data set, all with Percolator PEP—posterior error probability—of less than 0.001) was used to train the PredpI algorithm in the full pI range. After testing the performance of prediction with the new sets of pK constants using a test set (20,000 tightly focused peptides, different peptides than those in the training set but also with PEP < 0.001 and also from the A431 data set) the updated PredpI was employed in the creation of pI-restricted databases for the 6FT proteogenomics. The updated pK constants are supplied as Supplementary Data 1 and can be used with the PredpI algorithm previously published in Branca et al.[1].

**pI-restricted databases from 6FT of the human genome.** Human genome sequences build 37 (hg19) were downloaded from UCSC genome browser. Nucleotide sequences for each chromosome were in silico translated in six reading frames and in silico digested into peptides following trypsin rules (zero missed cleavages allowed, no cleaving on N-terminal side of proline residues). The python script (sixframetranslation.py) is available online at https://github.com/yafeng/proteogenomics_python. Peptide matches to known proteins were removed and unique peptides with length 8 aa to 30 aa were stored with their chromosome positions. The pI prediction algorithm, PredpI (now extended to the pH 3–10 range) was used to predict isoelectric points of all 6FT theoretical peptides, which were divided into pI-restricted databases with specific pI intervals corresponding to the experimental fractions of IPG strips. Due to both strip manufacturing and strip alignment variations during the process of extraction to 96-well micro-titer plate, the centers of pI intervals may shift slightly run-to-run and were therefore adjusted so that the median value of delta pI (experimental pI minus predicted pI) is equal to 0 for each individual IPG strip (the peptides used to calculate the delta pI shift were unique peptides identified with 0% FDR from the standard proteomics search). The pI interval of each pI-restricted database was extended on both sides of the experimental interval with the prediction error margin of ±0.11, which corresponds to the 95% confidence interval. Finally, each pI-restricted mini database was appended with Ensembl human protein database.

**Customized peptide database – human VarDB.** Human VarDB contains peptide sequences from pseudogenes, lncRNAs, nsSNPs, somatic mutations, and is concatenated with human known tryptic peptides (Supplementary Figure 18). Pseudogenes were downloaded from GENCODE[55] release 19 including also consensus pseudogenes predicted by the Yale and UCSC pipelines. Long non-coding RNAs were downloaded from both GENCODE release 19[56] and LNCipedia.org v 3.1[57]. These transcripts were translated in three reading frames and digested in silico into peptides (trypsin rules without missed cleavages). Redundant sequences matched

to known peptides were discarded. nsSNPs and somatic mutations were downloaded from CanProVar 2.0[58] and COSMIC release 71[59]. Proteins with substituted amino acid sequences were in silico digested to fully tryptic peptides (detailed steps of merging variant peptides from CanProVar and COSMIC are described in Supplementary Figure 19). The position of substituted amino acid in peptides was noted in order to run SpectrumAI.

**Proteogenomics search and class-specific FDR.** Peptide spectra were searched in two different databases in parallel: pI-restricted 6FT databases and VarDB. A separate target-decoy search strategy was used. Decoy peptides were generated from the peptides of corresponding target databases in reversed tryptic manner (i.e., C-terminal residue is maintained, whereas the rest of the target amino acid sequence is reversed). The VarDB search does not require knowledge of peptide pI and therefore can be broadly applied to any shotgun proteomics data set. A class-specific FDR[3] was estimated separately for novel and single amino acid variant (SAAV) peptides. Novel peptides are defined as sequences specifically from pseudogenes, lncRNA, or six-frame translations. SAAV peptides are defined as sequences specifically from CanProVar and COSMIC databases. First, target and decoy matches to known tryptic peptides were discarded (as well as deamidations of asparagine to aspartic acid and also considering that isoleucine=leucine). The FDR of novel/SAAV peptides was calculated as the number of decoy novel/SAAV peptides divided by the number of target novel/SAAV peptides above the score threshold. For the quantitative TMT analysis, novel peptide TMT reporter ion ratios were normalized per TMT channel using normalization factors based on median ratio centering calculated from the canonical protein tables of the standard proteomics searches.

**Mapping novel and SAAV peptides back to the genome.** The genomics coordinates of novel peptides identified from 6FT search were stored as peptide's ID at the six reading-frame translation step. It exempts the need to mapping them back to genome later. Novel peptides identified from VarDB search were mapped back to genome using genomic coordinates of their parent transcripts. The python script (map_novelpeptide2genome.py) to do this is posted online at https://github.com/yafeng/proteogenomics_python. SAAV peptides were mapped back to genome using coordinates annotated in dbSNP and COSMIC database. The python script (map_cosmic_snp_hg19.cor.py) is available at the above github address.

**SpectrumAI—automated spectral inspection.** The subset of novel and SAAV peptides with single amino acid substitution identified at 1% class-specific FDR were curated by SpectrumAI, which requires the peptides to fulfill two criteria. First, at least one of the peptide's MS2 spectra must contain ions flanking both sides of the substituted amino acid. For example, if a 12-amino-acid peptide is identified with single substitution at eighth residue, in order to pass SpectrumAI, it must have matched MS2 peaks (within 10 ppm fragment ion mass tolerance) from at least one of the following groups: $b_7$ and $b_8$, $y_4$ and $y_5$, $y_4$ and $b_7$, or $y_5$ and $b_8$. Second, the sum intensity of the supporting flanking MS2 ions must be larger than the median intensity of all fragmentation ions. An exception to these criteria is made when the substituted amino acid has a proline residue to its N-terminal side. Because CID/HCD fragmentation at the C-terminal side of a proline residue is thermodynamically unfavored, SpectrumAI only demands the presence of any b or y ions on the right (C-terminal) side of the substituted position. SpectrumAI is written in R and requires R libraries mzR[60] and MSnbase[61]. SpectrumAI and the R scripts used to generate MS2 spectrum mirror images (mirror plots) are deposited at: https://github.com/yafeng/SpectrumAI

**Comparisons with RNA-sequencing data.** RNA extraction and sequencing were done as part of and are detailed in Branca et al.[1] and Uhlen et al.[62]. Briefly, total RNA was prepared using the RNeasy Mini kit (Qiagen) according to the manufacturer's instructions. The experion automated electrophoresis system (Bio-Rad) was used to assess RNA quality. Sequencing was performed on HiSeq2000 (Illumina) with the standard Illumina RNA-seq protocol. RNA-sequencing data can provide complementary mRNA-level evidence for the expression of a putative novel peptide. We consider an aligned RNA sequencing read to be a match to a genomic region corresponding to a detected peptide if the read: (1) is uniquely aligned to the locus (or part of the locus), (2) has at-most one mismatch to the reference in the peptide region itself, and (3) is properly paired in cases where paired-end sequencing was performed. Reads flagged as multi-mapping would thus not count as mRNA evidence for the peptide; neither would spliced alignments that contain genomic regions adjacent to the peptide locus but not the locus itself. We developed a Python script for counting, from a set of BAM files and a GFF3 format file of identified peptides, how many mapped RNA-seq reads fulfilling the above criteria overlapped each peptide. The criteria are adjustable in the script. The script is available at: https://github.com/yafeng/proteogenomics_python/scam_bams.py

To generate BAM files as input to this script, we used STAR[63] to align samples from the A431 cell line and four human tissues (we did not have RNAseq data for the placenta samples) to the hg19 version of the human genome assembly.

**Variant analysis.** Whole genome sequencing data on A431 cells from Akan et al.[64] was downloaded (available in SRA ERP001947). FASTQ files were mapped to

GRCh38 using bwa mem[65] (version 0.7.13-r1126). The resulting sam file was converted to bam, sorted, and indexed using samtools[66] (Version 1.3.1). A431 RNA-seq data (uploaded to ArrayExpress with ID E-MTAB-5285) were mapped against GRCh38 using STAR[63] (Version STAR_2.5.2a). Resulting bam files were treated with sambamba[67] to expand intron gaps and fed to FreeBayes, version v1.0.2: https://arxiv.org/abs/1207.3907

Variants in the resulting index/sorted bam file were called using FreeBayes (version v1.0.2, option –C 5, the rest were default). All tools and reference data were downloaded and compiled/configured using bcbio. The coordinates of missense variants were converted to hg19 coordinates to compare with those of missense variants identified at peptide level.

**Evolutionary conservation**. We assessed the evolutionary conservation of the genomic regions encoding the peptides by calculating mean scores for each region based on the PhastCons[68] 100-way vertebrate multiple alignment tracks available from: http://hgdownload.cse.ucsc.edu/goldenPath/hg19/phastCons100way/hg19.100way.phastCons.bw

The calculations were done using a python script and posted at: https://github.com/yafeng/proteogenomics_python/calculate_phastcons.py

The distribution of mean conservations scores for the peptide regions were compared with randomly selected sets of pseudogenes from the pseudogene.org database and to a high-confidence set of lncRNAs from lncipedia.org. Links here: http://www.pseudogenes.org/psidr/data/gencode.v7.pseudogene.txt http://www.lncipedia.org/downloads/lncipedia_3_1_hc.gtf

**Protein-coding potential based on conservation**. We used PhyloCSF[34] results to determine the protein-coding potential of the putative novel peptides. A Python script was used to parse the PhyloCSF bigWig files (https://data.broadinstitute.org/compbio1/PhyloCSFtracks/hg19/latest/PhyloCSF) in order to classify each peptide region as "coding", "non-coding", or "no-call" based on the PhyloCSF coding potential scores for each region (in all six reading frames). Finally, peptides that had been classified as "coding" in at least one reading frame were selected as putative protein coding. The python script is posted at: https://github.com/yafeng/proteogenomics_python/calculate_phylocsf.py

**Transcription and translation evidence for new coding loci**. To find additional levels of evidence for the discovered coding loci, we searched for evidence in previously published ribosome profiling studies and transcriptional gene expression profile by CAGE (Cap Analysis Gene Expression) data. The former indicates evidence for translation initiation sites (TIS), the latter for transcription start sites (TSS). Bigwig tracks of ribosome profiles of THP-1 cell lines published by Fritsch et al.[30] and mapped CAGE reads across a panel of biological samples published in Forrest et al.[32] were downloaded from UCSC genome browser. 10000 random genomic intervals (with same length distribution as that of our novel peptides) were generated to estimate how many CAGE and Ribo-seq reads randomly map to genomic loci by chance (Supplementary Figure 20).

**Code availability**. The automated IPAW pipeline and user manual is available at: https://github.com/lehtiolab/proteogenomics-analysis-workflow

Individual python scripts used in the proteogenomics workflow are deposited in: https://github.com/yafeng/proteogenomics_python

SpectrumAI R code is deposited in: https://github.com/yafeng/SpectrumAI

**Data availability**. The proteomics data of A431 cells and normal tissues have been deposited to the ProteomeXchange Consortium via the PRIDE partner repository with the data set identifier PXD006291 (https://www.ebi.ac.uk/pride/archive/projects/PXD006291).

RNA-seq data of A431 cells after gefitinib treatment is available at ArrayExpress with accession ID E-MTAB-5285 (https://www.ebi.ac.uk/arrayexpress/experiments/E-MTAB-5285/). RNA-seq data of normal tissues (kidney, liver, tonsil, testis but not placenta) was originally collected for the work of Uhlen et al.[62] and is available via the ID E-MTAB-2836 at ArrayExpress (https://www.ebi.ac.uk/arrayexpress/experiments/E-MTAB-2836/).

All other data supporting the findings of this study are available from the corresponding authors on reasonable request.

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

## Acknowledgements

Anna Asplund and Fredrik Pontén (Uppsala University, Sweden) as well as Mathias Uhlen and Björn Hallström (Royal Institutet of Technology, KTH, Sweden) are gratefully acknowledged for providing tissue material for the study. Support by NBIS (National Bioinformatics Infrastructure Sweden) is gratefully acknowledged. Parts of the computations were performed on resources provided by SNIC through Uppsala Multi-disciplinary Center for Advanced Computational Science (UPPMAX). Funding from the Swedish Foundation for Strategic Research, Swedish Cancer Society, Swedish Research Council, Swedish Childhood Cancer Foundation, The Cancer Research Funds of Radiumhemmet, Stockholm's County Council (ALF funding), and EU FP7 project GlycoHit is gratefully acknowledged.

## Author contributions

J.L., R.M.M.B., L.M.O., and H.J.J. conceived and designed the experiments. R.M.M.B., H.J.J., and M.V. performed the biological sample preparation. R.M.M.B. and H.J.J. performed the HiRIEF peptide-level separation and the LC-MS analysis. Y.Z., R.M.M.B., L.M.O., H.J.J., and J.L. designed the proteogenomics workflow. Y.Z. wrote the SpectrumAI script following suggestion from H.J.J. Y.Z. wrote the proteogenomics scripts with advice from J.B., M.H., and A.F.-W. M.H. wrote the python scripts for searching orthogonal evidence. J.B. integrated the proteogenomics workflow into nextflow with advice from Y. Z. Y.Z. did the single amino acid variant analysis. R.M.M.B. and Y.Z. did the discovered protein-coding loci analysis. M.H., Y.Z., and A.F.-W. did the orthogonal validation analysis. Y.Z. and R.M.M.B. wrote the manuscript with assistance from L.M.O. and J.L. All authors were involved in the discussion of the manuscript and approved its final version.
