## [Peer Review File · Nature Communications]

REVIEWERS' COMMENTS:

Reviewer #1 (Remarks to the Author):

The team have covered all my point well and I'm happy for the manuscript to be published.

Reviewer #2 (Remarks to the Author):

This is a revised version of the manuscript that presents a comprehensive proteogenomics pipeline. The authors made a good effort trying to address my concerns, and the manuscript has improved. I am still concerned about the number of novel identifications reported and their nature (pseudogenes, lncRNA, etc.). There is so much controversy especially about the lncRNA expression at the protein level, and many do not believe in their protein expression. However, I think the authors did a good job trying to convince themselves and the readers that the analysis is solid, at least in that it addresses the common proteogenomics pitfalls. Because they make all their data available on ProteomeXchange, a larger computational community can mine these data in the further confirm or dis-confirm their findings. Thus, I am supportive of the publication of this paper in Nature Communications.

Reviewer #3 (Remarks to the Author):

I appreciate the authors' efforts to improve the manuscript. Despite the fact that I like to see the publication of such work in the field, I unfortunately hesitate to give my full support to the current version of the manuscript, for the following reasons:

1. The authors claim that this is a methods paper, but I am still not convinced that the individual methods that they have developed are novel enough. Indeed, the proposed proteogenomic workflow – which integrates discovery, curation and validation steps – is not trivial at all and should be highly appreciated in the field. Unfortunately the workflow itself does not function as a whole; rather, it consists of several separate (Python) scripts. It presents significant difficulty for readers, especially for those who are not experts in Python, to understand how exactly the authors did the work. I would therefore advise the authors to include a step-by-step protocol that describes explicitly how to make use of each one of the scripts, and how to put them all together as a comprehensive working pipeline.
2. The title does not reflect the content of the manuscript. It suggests an article that contains a systematically quantitative analysis of novel non-coding regions in different tissues. However, in fact it focusses more on the development of the workflow, which does not make use of any quantitative information at all. The authors did not even systematically reveal and assess the whole quantitative results, but merely picked out a few cases for demonstration, without any attempt at independent biological validation. I would advise the authors to change the title and, again (as this is a methods paper), to include the quantification assessment/analysis in their proposed validation workflow, such as overall quantitative reproducibility in replicates, general variation, and quantitative distribution of normal peptides versus novel peptides, etc.
3. Page 14 line 345: "Out of these 100 synthetic peptides, 2 failed to generate good fragmentation spectra, 7 demonstrated their endogenous counterparts to be incorrect identifications upon manual inspection...". Does that mean that even after careful curation and validation there were still around 7–9% of false positives in the data set? It might enforce the importance of the curation/validation steps in proteogenomics, as the authors proposed. However, is there any way to improve it further? The authors should comment on this aspect in the manuscript.

ANSWERS TO REVIEWERS' COMMENTS:

We thank all the reviewers for the thorough reading of our manuscript and all the constructive criticism, which we feel has led to a substantial improvement of our work. Detailed answers are below.

Reviewer #1 (Remarks to the Author):

The team have covered all my point well and I'm happy for the manuscript to be published.

Reviewer #2 (Remarks to the Author):

This a revised version of the manuscript that presents a comprehensive proteogenomics pipeline. The authors made a good effort trying to address my concerns, and the manuscript has improved. I am still concerned about the number of novel identifications reported and their nature (pseudogenes, lncRNA, etc.). There is so much controversy especially about the lncRNA expression at the protein level, and many do not believe in their protein expression. However, I think the authors did a good job trying to convince themselves and the readers that the analysis is solid, at least in that it addresses the common proteogenomics pitfalls. Because they make all their data available on ProteomeXchange, a larger computational community can mine these data in the further confirm or dis-confirm their findings. Thus, I am supportive of the publication of this paper in Nature Communications.

Reviewer #3 (Remarks to the Author):

I appreciate the authors' efforts to improve the manuscript. Despite the fact that I like to see the publication of such work in the field, I unfortunately hesitate to give my full support to the current version of the manuscript, for the following reasons:

1. The authors claim that this is a methods paper, but I am still not convinced that the individual methods that they have developed are novel enough. Indeed, the proposed proteogenomic workflow – which integrates discovery, curation and validation steps – is not trivial at all and should be highly appreciated in the field. Unfortunately the workflow itself does not function as a whole; rather, it consists of several separate (Python) scripts. It presents significant difficulty for readers, especially for those who are not experts in Python, to understand how exactly the authors did the work. I would therefore advise the authors to include a step-by-step protocol that describes explicitly how to make use of each one of the scripts, and how to put them all together as a comprehensive working pipeline.
2. The title does not reflect the content of the manuscript. It suggests an article that contains a systematically quantitative analysis of novel non-coding regions in different tissues. However, in fact it focusses more on the development of the workflow, which does not make use of any quantitative information at all. The authors did not even systematically reveal and assess the whole quantitative results, but merely picked out a few cases for demonstration, without any attempt at independent biological validation. I would advise the authors to change the title and, again (as this is a methods paper), to include the quantification assessment/analysis in their proposed validation workflow, such

as overall quantitative reproducibility in replicates, general variation, and quantitative distribution of normal peptides versus novel peptides, etc.

3. Page 14 line 345: “Out of these 100 synthetic peptides, 2 failed to generate good fragmentation spectra, 7 demonstrated their endogenous counterparts to be incorrect identifications upon manual inspection...”. Does that mean that even after careful curation and validation there were still around 7–9% of false positives in the data set? It might enforce the importance of the curation/validation steps in proteogenomics, as the authors proposed. However, is there any way to improve it further? The authors should comment on this aspect in the manuscript.

1. We acknowledge that the usage of the several scripts was far from trivial. We have thus worked on improving the workflow by consolidating all the steps into what now simply runs as a one-liner command. The automated IPAW pipeline and user manual is now available at github. See “Code availability” at the end of the Methods section. We believe that this improvement has greatly increased the user friendliness of the workflow.
2. We realize that the reviewer is correct in that the title of the manuscript was not adequate. We have thus changed the title to “Discovery of coding regions in the human genome by Integrated Proteogenomics Analysis Workflow.”
3. Indeed the intention of reporting that result from the synthetic peptide analysis was to highlight the difficulty/importance of controlling false positives in findings in the proteogenomics field. We have added the following sentence in the results section commenting on this matter: “The significant number (7 out of 98) of incorrect novel peptides highlights the need for further efforts into curation/validation of findings in the proteogenomics field.”